# Monitoring of the Organophosphate Pesticide Chlorpyrifos in Vegetable Samples from Local Markets in Northern Thailand by Developed Immunoassay

**DOI:** 10.3390/ijerph17134723

**Published:** 2020-06-30

**Authors:** Surat Hongsibsong, Tippawan Prapamontol, Ting Xu, Bruce D. Hammock, Hong Wang, Zi-Jian Chen, Zhen-Lin Xu

**Affiliations:** 1Research Institute for Health Science, Chiang Mai University, Chiang Mai 50200, Thailand; tippawan.prapamontol@cmu.ac.th; 2Beijing Key Laboratory of Biodiversity and Organic Farming, College of Resources and Environmental Sciences, China Agricultural University, Beijing 100193, China; xuting@cau.edu.cn; 3Department of Entomology and Nematology and UCD Comprehensive Cancer Center, University of California Davis, Davis, CA 95616, USA; bdhammock@ucdavis.edu; 4Guangdong Provincial Key Laboratory of Food Quality and Safety, College of Food Science, South China Agricultural University, Guangzhou 510642, China; gzwhongd@163.com (H.W.); czj1q2w3e4r5t@163.com (Z.-J.C.); jallent@163.com (Z.-L.X.)

**Keywords:** chlorpyrifos residue, ic-ELISA, GC-FPD, northern Thailand

## Abstract

Chlorpyrifos is an organophosphate pesticide that is wildly used among farmers for crop protection. However, there are concerns regarding its contamination in the environment and food chain. In the present study, an in-house indirect competitive enzyme-linked immunosorbent assay (ic-ELISA) specific for detecting chlorpyrifos is developed and validated against gas chromatography–flame photometric detection (GC-FPD) as the conventional method. The developed ic-ELISA was used for detecting chlorpyrifos residue in vegetable samples. The developed ic-ELISA showed good sensitivity to chlorpyrifos at an IC_50_ of 0.80 µg/kg, with low cross-reactivity to other organophosphate pesticides. The 160 samples were collected from local markets located in the Chiang Rai, Chiang Mai, and Nan provinces in northern Thailand. The positive rate of chlorpyrifos residues in the vegetable samples was 33.8%, with the highest levels found in cucumbers, coriander, and morning glory, at 275, 145, and 35.3 µg/kg, respectively. The highest median levels of chlorpyrifos found in the detected samples were Chinese cabbage (332 μg/kg), cucumber (146.3 μg/kg) and Chinese Kale (26.95 μg/kg). The developed ic-ELISA is suitable for the rapid quantitation of chlorpyrifos residues.

## 1. Introduction

Chlorpyrifos (O,O-diethyl O-(3,5,6-trichloro-2-pyridinyl) phosphorothioate) is a broad-spectrum organophosphate pesticide, which is the most popular among farmers for protecting their crops by controlling different kinds of pests [1]. It is extremely toxic to a wide range of nontarget aquatic organisms [2]. Additionally, the consumer’s health risk is a concern due to presence of residues in fruits and vegetables [3,4], the environment [5,6], and all of the agricultural products [7,8,9]. This is possible because farmers use high volumes of chlorpyrifos, and thus it has a chance to be detected in many samples and agricultural production in general. Chlorpyrifos can inhibit cholinesterase, i.e., acetyl cholinesterase and butyryl cholinesterase [10,11], and can be cause damage to the central nervous system [12], as well as damage to the sympathetic and parasympathetic nervous systems [13]. Moreover, in human epidemiological studies, occupational exposure to chlorpyrifos has been related to neurological and neuro-behavioral deficits, including cognitive impairment [14,15,16,17,18]. Chlorpyrifos also has been shown to produce learning deficits in rats after acute and repeated administration, similar to those induced in Alzheimer’s disease (AD) [19]. Therefore, chlorpyrifos has been listed as banned in several countries, including Thailand. However, monitoring still needs to be carried out given its illegal use, because it is a very toxic substance.

Currently, most of the techniques and standard methods for detecting chlorpyrifos are chromatographic based-methods, i.e., liquid chromatography [20], gas chromatography [8,21,22], and mass spectrometric methods [23,24,25]. Chromatographic-based methods have complicated steps, require skilled technical help, are equipment intensive, require a skilled person, and require a significant amount of time. Therefore, a low number of samples can be analyzed per day, and thus a high number of samples cannot be processed in a short time. Additionally, it also very expensive per sample. Thus, the immunoassay complements instrumental methods in many ways, in particular being more applicable in handling a higher number of samples and field assays [26,27,28,29,30]. The absorbance or color are inversely proportional to compounds in the competitive immunoassay. It is useful to use immunoassay as a tool for detecting chlorpyrifos in a high number of samples for epidemiology studies.

This paper reports on a developed immunoassay for detecting the organophosphate pesticide chlorpyrifos based on mouse polyclonal antibody and its application for the monitoring of chlorpyrifos residues in vegetable samples from the Chiang Mai, Chiang Rai, and Nan provinces in the northern part of Thailand. Analytical parameters, such as the sensitivity, accuracy, and reproducibility, were evaluated, and the results are discussed in comparison with GC-FPD. The residues in vegetable samples were analyzed by using the developed immunoassay as an analytical method.

## 2. Materials and Methods

### 2.1. Chemicals and Reagents

The 3-mercaptopropionic acid, potassium hydroxide (KOH), potassium iodide (KI) hydrochloric acid (HCl), sodium hydroxide (NaOH), bovine serum albumin (BSA), and ovalbumin (OVA) were obtained from Sigma-Aldrich Chemie (Steinheim, Germany). The Bio-Rad protein assay kit was from, BioRad (California, USA). Sodium chloride, tetrabutylammonium iodide (TBAI), disodium hydrogen phosphate, sodium dihydrogen phosphate, potassium dihydrogen phosphate, potassium chloride, sodium carbonate, and citric acid were from Merck (Hohenbrunn, Germany). The organic solvents, i.e., acetonitrile, dimethylformamide, methanol (MeOH), and ethyl acetate, were from J.T. Baker (Pennsylvania, USA). The 3,3′,5,5′-tetramethylbenzidine (TMB) was from Fluka (Steinheim, Germany), and the tetrahydrofuran (THF), methanol (MeOH) dimethyl formamide (DMF), and dimethyl sulfoxide (DMSO) was from Sigma-Aldrich (Steinheim, Germany). Goat anti-mouse IgG (H+L) was from Invitrogen (Rockford, USA).

### 2.2. Standard Organophosphate Pesticides

Eighteen organophosphate pesticide (OPs) standards (purity 93–99%) were purchased from Dr. Ehrenstorfer (Augsburg, Germany) and used for cross-reactivity analysis by immunoassay and analysis by gas chromatography–flame photometric detection (GC-FPD), i.e., chlorpyrifos-methyl (purity 98.7%), dichlorvos (purity 98.7%), mevinphos (purity 96.9%), omethoate (purity 93.4%), dicrotophos (purity 97.5%), monocrotophos (purity 98%), dimethoate (purity 99.3%), diazinon (purity 98.9%), parathion-methyl (purity 98.2%), fenitrothion (purity 98.6%), malathion (purity 99.4%), chlorpyrifos (purity 99%), primiphos-ethyl (purity 99%), methidathion (purity 98.3%), prothiophos (purity 95.4%), profenofos (purity 99%), ethion (purity 98.8%), triazophos (purity 98.9%), ethyl4-nitrophenyl phenylphosphonothioate (EPN) (purity 98.2%), azinphos-ethyl (purity 99.4%), and azinphos-methyl (purity 99.6%). The standard pesticides were prepared in methanol and diluted in 20% methanol in PBS (pH 7.2) before being analyzed by ic-ELISA, and was also stored at −20 °C prior to the analysis.

### 2.3. Development of Immunoassay for Detecting Chlorpyrifos

The immunoassay was developed in accordance to a similar protocol as in previous studies, such as a hapten synthesis, preparation of immunogens and coating antigens, characterizing the polyclonal antibody, and the matrix effect of the sample extract on the antibody [31,32,33,34].

#### 2.3.1. Hapten Synthesis

The hapten was synthesized by following a previously published method [35,36], with some modification. The haptens were used for preparation of the immunogens and coating antigens as well as the developed immunoassay. The haptens were synthesized as follows in Figure 1:

The structures of the haptens are shown in Figure 1. Synthetic intermediates were analyzed by thin layer chromatography (TLC) and confirmed by ^1^H- and ^13^C-nuclear magnetic resonance (NMR).

(1)*Hapten (a) and (b)*: To a solution of chlorpyrifos (1.00 g, 2.85 mmol) dissolved in ethanol (5 mL), KOH (400 mg, 7.13 mmol) was added and stirred for 10 min. The resulting solution was treated with 3-mercaptopropanoic acid (0.30 mL, 3.45 mmol), and the temperature was raised to 80 °C and refluxed for 1 h. The solvent was removed under reduced pressure, followed by washing three times with hexane (20 mL each time). The residue was acidified to a pH of 2 and extracted with 20 mL of dichloromethane three times. The organic phase was dried over sodium sulfate and concentrated under reduced pressure to give the crude product, which was purified by silica gel column chromatography. The product was mixed between (a) and (b) in a 2:1 ratio with an 83% yield (0.68 g). Hapten (a), ^1^H NMR (CDCl_3_, 400 MHz) δ 7.65 (s, 1H), 4.33 (m, 4H), 3.41 (t, J = 7.2 Hz, 2H), 2.90 (t, J = 7.2 Hz, 2H), 1.41 (t, J = 7.2 Hz, 6H); ^13^C NMR (CDCl_3_, 100 MHz) δ 176.6, 153.6, 151.4, 138.9, 125.0, 116.0, 66.4, 66.3, 34.1, 25.3, 15.9, 15.8 and hapten (b), ^1^H NMR (CDCl_3_, 400 MHz) δ 7.65 (s, 1H), 4.33 (m, 4H), 3.41 (t, J = 7.2 Hz, 2H), 2.90 (t, J = 7.2 Hz, 2H), 1.41 (t, J = 7.2 Hz, 6H); ^13^C NMR (CDCl_3_, 100 MHz) δ 176.6, 153.6, 151.4, 138.9, 125.0, 116.0, 66.4, 66.3, 34.1, 25.3, 15.9, 15.8.(2)*Hapten (c)*: To a solution of chlorpyrifos (1.00 g, 2.85 mmol) dissolved in 3:2:1 THF/MeOH/water (5 mL), LiOH (82 mg, 3.42 mmol) was added and stirred for 30 min. The resulting solution was evaporated before adding DMF and then treated with methyl-6-bromohexanoate (715 mg, 3.42 mmol) and a catalytic amount of TBAI and KI. After refluxing for 1 h, the reaction mixture was allowed to cool down to room temperature before further hydrolysis with NaOH. After completion of the reaction, the residue was acidified to a pH of 2 and extracted with 20 mL of dichloromethane three times. The extraction phase was dried over sodium sulfate and concentrated under reduced pressure to give the crude product, which was purified by silica gel column chromatography. The product was obtained with a 69% yield (0.62 g). ^1^H NMR (CDCl_3_) δ 7.74 (s, 1H, ArH), 4.92 (s, 2H, OCH_2_); ^13^C NMR (CDCl_3_, 100 MHz) δ 171.0, 158.6, 145.7, 140.2, 127.2, 120.7, 65.6.

#### 2.3.2. Preparation of the Immunogen and Coating Antigen

The mixed carboxylic acid hapten was attached to both a protein and an enzyme using the active ester method [37]. Hapten was conjugated to bovine serum albumin (BSA) for immunogen preparation and conjugated to oval albumin (OVA) for coating antigen preparation. The hapten densities of the immunogen and coating antigen were estimated based on mass spectra of ultraviolet spectra to confirm coupling and then calculated [38]. The synthesized immunogen and coating antigen demonstrated qualitative differences between the corresponding carrier protein and conjugate in the UV–Vis spectra, indicating that the hapten had been conjugated to the carrier protein successfully. The estimation of molar absorptivity of the hapten was the same for the free and conjugated forms of the carrier protein. The estimated molar ratio of the immunogen and coating antigen were 12 and 22, respectively. The protein contents of the hapten–protein conjugates in the dialysates were determined according to the Bio-Rad dye based on the Bradford protein assay.

#### 2.3.3. Preparation of Polyclonal Antibody (pAb)

The animals were reared in a clean, standard environment, with a food and water supply. The immunization experimental protocol with animals was performed in accordance with relevant institutional and national guidelines and regulations, and it was approved by the Animal Care and Use Committee, Chiang Mai University (Protocol number: 2561/MC-004). Two BALB/c mice (age 6–8 weeks) were immunized so their immune system would produce polyclonal antibodies by using the prepared immunogen for induction of antibody responses against hapten according to previous studies [38,39]. Two mice were inoculated with 100 µg of immunogen dissolved in 10 mM phosphate buffer saline (PBS) and then emulsified with complete Freund’s adjuvant (1:1, v/v) injected through the subcutaneous (s.c.) route at multiple sites on the back. Then, they were given three booster immunizations with 100 µg of immunogen dissolved in PBS and then emulsified with incomplete Freud’s adjuvant (1:1, v/v) with substitution with Freund’s complete adjuvant every two weeks in the same way. Mice were bled from the tail veins before the immunization schedule first and then every week. The serum was separated from the blood by centrifugation and stored in the freezer prior to use for evaluation of the antibody response against hapten by direct ELISA using hapten-OVA as the coating antigen. The serum from a sensitive mouse from the 3rd to 5th immunization were collected from the tail (0.3 mL each time). After the 6th immunization for 3 days, the heart blood of the mice was collected. Mice were placed in a deep sleep by saturated carbon dioxide before heart blood collection. All the collected serums were mixed and the pooled serum from the heart blood was prepared (the reciprocal titer about 51,200) and used for further experiments.

#### 2.3.4. Development of ic-ELISA

(1)*Methanol effect:* Methanol is a good solvent for immunoassays and has been used in many previous studies [40,41,42]. However, the methanol content in PBS may affect the antibody. The methanol contents in PBS were studied by using different concentrations of methanol, i.e., 50%, 40%, 20%, 10%, and 5%, in PBS as a diluent for chlorpyrifos in several concentrations. The absorbance of each methanol content and IC_50_ were compared, and due to the good results and no effect from methanol, it was selected as the diluent of the developed immunoassay.(2)*Ionic strength:* The ionic strength affected the ic-ELISA, and thus the standard curves of chlorpyrifos were analyzed by using different concentrations of 10 mM PBS at a pH of 7.0, i.e., 1x, 2x, 3x, 4x, 5x, and DI water.(3)*Indirect Competitive ELISA (ic-ELISA):* The ic-ELISA was performed according to the method of Hongsibsong et al. [39]. The concentrations of antibody and coating antigen were optimized by checkerboard titration. The good condition was coating the antigen at 1 µg/mL and a serum dilution at 1:1000. The ic-ELISA was performed by using the optimal concentration as follows. Microtiter plates (Maxisorb, NUNC, Roskilde, Denmark) were coated with 100 µL/well of the hapten-OVA (1 µg/mL) as a coating antigen in a carbonate buffer at a pH of 9.6 and allowed to sit overnight at 4 °C. The plates were washed with PBS plus 0.05% Tween 20 (PBST) and blocked with 200 µL/well of 1% (w/v) gelatin in PBS at a pH of 7.2. After 1 h of incubation at room temperature, the plates were washed as described previously. Standards (or samples extracted) were mixed with equal volumes of serum diluted in PBS (1:1000) and pre-incubated for 1 h at room temperature. The pre-incubated mixture was transferred to the wells (100 µL/well) and incubated for 1 h at room temperature for competition. Then, the plate was washed by PBST, and 100 µL/well of 1:5000 HRP conjugated goat anti-mouse IgG (H+L) in PBS at a pH of 7.2 was added. After 1 h, the plate was washed, and 100 µL of a substrate solution (0.1 mL of 1% H_2_O_2_ and 0.4 mL of 0.6% 3,3′,5,5′-tetramethylbenzidine in dimethyl sulfoxide (DMSO) were added to 25 mL of citrate-acetate buffer, pH = 9.6) was added to each well. The plates were stopped with 50 μL of 2N H_2_SO_4_ and read by an ELISA plate reader (Sunrise, Salzburg, Austria) at 450 nm. The development of a yellow color was inversely proportional to the amount of chlorpyrifos present. The absorbance was calculated for 50% inhibition by a nonlinear curve fit. The concentration of chlorpyrifos residue was calculated from the standard curve.(4)*Cross-reactivity (CR):* The cross-reactivity was studied by ic-ELISA and substitution of the chlorpyrifos standard or sample with an organophosphate pesticide in the same group as chlorpyrifos. Organophosphate pesticide standards were used for cross-reactivity by immunoassay, i.e., chlorpyrifos-methyl, dichlorvos, mevinphos, omethoate, dicrotophos, monocrotophos, dimethoate, diazinon, parathion-methyl, fenitrothion, malathion, chlorpyrifos, primiphos-ethyl, methidathion, prothiophos, profenofos, ethion, triazophos, ethyl 4-nitrophenyl phenylphosphonothioate (EPN), azinphos-ethyl, and azinphos-methyl. The cross-reactivity was determined according to the equation below:

CR (%) = (IC_50_ (chlorpyrifos) / IC_50_ (interferent)) × 100.(1)

(5)*Matrix effect of the color from the vegetable sample:* Since vegetables have colors, the effects of the various colors of vegetables on the antibody were studied. The green (kale), red (tomato), and white (Chinese cabbage) colors, which are commonly consumed in the Thai community, were evaluated. The vegetable samples were chopped into small pieces and extracted following a previously described method [4,43]. The best methanol content was used and the effect of the extraction of each colored vegetable on the antibody was determined. The recovery was computed by spiking chlorpyrifos into a pooled vegetable sample and then extracting before analyzing. The pooled vegetable was prepared from those three kinds of vegetables with no chlorpyrifos residues after analyzing by GC-FPD.(6)*Immunoassay validation:* To evaluate the performance of the developed ic-ELISA for organophosphate pesticide chlorpyrifos, three experiments were performed: (1) the recoveries of spiked pooled vegetable samples were measured by ic-ELISA, and the accuracy, precision, limit of detection (LoD), and limit of quantification (LoQ) were reported as the percent of recovery, percent of coefficient of variance (%CV), IC_15_, and IC_20_, respectively; (2) the applicability of ic-ELISA to detect chlorpyrifos was determined by analyzing 70 blind vegetable samples, and the results were compared against the results from GC-FPD, which is the standard technique with a specific detector for analyzing compounds that include phosphate in their molecule; and (3) the application of the developed ic-ELISA to analyze the organophosphate pesticide chlorpyrifos in vegetable samples. The vegetable samples, i.e., coriander (*n* = 28), yard long bean (*n* = 23), cabbage (*n* = 17), pakchoi (*n* = 17), Chinese cabbage (*n* = 13), morning glory (*n* = 13), cauliflower (*n* = 9), spring onion (*n* = 8), broccoli (*n* = 7), chili (*n* = 7), eggplant (*n* = 6), Chinese kale (*n* = 5), cucumber (*n* = 4), and tomato (*n* = 3), were collected from local markets in the Maetang district, Chiang Mai province (*n* = 28); Phaya Mengrai district, Chiang Rai province (*n* = 77); and Muang district, Nan province (*n* = 55)—for a total of 160 samples. Five kilograms of each sample was collected in each market; the edible parts were finely chopped and 300 g was randomly taken for analysis. Then, all of the samples were transferred to the Toxicology Laboratory, Environmental and Health Research Unit, Research Institute for Health Science, Chiang Mai University, and kept in a −20 °C freezer prior to analysis.

#### 2.3.5. Analysis of the Organophosphate Pesticide Residues in the Vegetable Samples

(1)*Sample extraction and clean up:* The sample extraction followed the method of [4,39]. Briefly, 5 g of vegetable sample was weighed into a 50 mL centrifuge tube, followed by the addition of 10 mL of acetonitrile (high-performance liquid chromatography grade). Two hundred fifty microliters of 5 μg/mL triphenylphosphate (internal standard (IS)) was added and subsequently centrifuged for 5 min at 2500 rpm. The supernatant was transferred to a 50 mL centrifuge tube with the addition of 6 g of MgSO_4_ and 3 g of NaCl, followed by centrifugation again for 5 min at 2500 rpm. The extract was evaporated to complete dryness using a vacuum rotary evaporator (Buchi, Flawil, Switzerland) with a water bath at 30 to 35 °C and then reconstituted with 5 mL of ethyl acetate. One milliliter of ethyl acetate phase was pipetted into 2 dispersive solid-phase extraction tubes and centrifuged for 3 min at 2000 rpm. Finally, the extract was evaporated with a gentle stream of nitrogen at room temperature and subsequently reconstituted in 1.0 mL of 10% methanol in PBS at a pH of 7.0 for the immunoassay, and another one was reconstituted with 0.5 mL of ethyl acetate for gas chromatography (GC) analysis.(2)*Gas chromatography–flame photometric detection (GC-FPD):* A Hewlett-Packard model 6890 equipped with a flame photometric detector, a capillary column (DB-5MS, 0.25 mm × I.D. × 30 m length × 0.25 μm film thickness (Agilent J & W column; Agilent Technologies, DE, USA), and a computerized data handling system (GC Chemstation A.10.02; Agilent Technologies, CA, USA) was used. The temperature was 220 °C for the injection port (splitless mode). The temperature programming of the oven was as follows: initial temperature of 100 °C for 10 min, first ramp at 15 °C/min to 180 °C (5 min), second ramp at 5 °C/min to 250 °C (3 min), and the final temperature maintained at 290 °C for 4 min. The carrier gas was 99.999% helium.

## 3. Results and Discussion

### 3.1. Development of the Immunoassay

Since chlorpyrifos (chlorpyrifos-ethyl) has been imported to Thailand, and in a previous study shown to be the most detected residue in vegetable samples [4], a simpler method was required for detecting chlorpyrifos in a high number of samples. This study developed an in-house immunoassay and applied it for detecting chlorpyrifos in vegetable samples. The hapten was synthesized by following methods previously published [35,36]. It was confirmed that the procedure for preparing the hapten could be based on a previous study [35,36], which published a structure of hapten (hapten (a)) that could be used to produce the antibody to chlorpyrifos. The coating antigen prepared by using the same hapten with immunogen did not give good sensitivity for ic-ELISA (data not shown). The polyclonal antibody in the serum from a mouse was used for developing the immunoassay by using the pooled serum from after the 3rd immunization of heart blood (3.0 mL) because the titer of antibody in the serum samples was equal and used for analyzing the vegetable samples. The dilution of serum at 1:1000 was used for the determination of chlorpyrifos and the validation of the ic-ELISA, along with a good coating antigen, namely, (c)-OVA (compound (c) in Figure 1), which gave the lowest concentration inhibition of chlorpyrifos at an IC_50_ of 0.80 µg/kg.

Sensitivity and specificity based on the optimized ic-ELISA and the standard curves of chlorpyrifos were developed. The developed ic-ELISA showed high sensitivity to chlorpyrifos with an IC_50_ = 0.80 ± 0.56 μg/kg. The cross-reactivities (CRs) with dichlorvos, mevinphos, omethoate, dicrotophos, monocrotophos, dimethoate, diazinon, parathion-methyl, fenitrothion, malathion, primiphos-ethyl, prothiophos, ethion, triazophos, ethyl 4-nitrophenyl phenylphosphonothioate (EPN), azinphos-ethyl, and azinphos-methyl at very high concentrations were minimal (>1000 µg/kg). The pAb gave a cross-reactivity with chlorpyrifos-methyl, profenofos, and methidathion at 125%, 14.28%, and 0.28%, respectively. Chlorpyrifos can be determined at concentrations ranging from 20.0 to 2090 µg/kg. The IC_50_ of the present study had both higher [44,45] and lower [45,46] values than previously reported. The specificity of the produced pAb was good for detecting chlorpyrifos in the sample because it had low cross-reactivity to other OPs, except for chlorpyrifos-methyl, which has no reported use in Thailand (the Office of Agricultural Regulation), and pAb can bind the group of organophosphate pesticides in a smaller number with a high concentration. The hapten with a similar structure to that in the present study gave the same results [45], and this was helpful for developing methods for detecting chlorpyrifos in real samples.

The matrix effect is one of the main factors that restricts the application of immunoassays in pesticide analysis [32,33,47]. In the development of immunoassays for pesticides in vegetables samples, the direct extraction of pesticides from food samples with organic solvents, such as methanol, is commonly used [40,41,42]. In practice, a dilution of 20–100 times of the methanol extraction should be done to reduce the matrix effect prior to immunoassays. However, the dilution can cause a reduction in the assay sensitivity, and an overlarge dilution would cause inapplicability of the immunoassays. To confirm the applicability based on the developed ic-ELISA, different colored vegetable samples were used as matrix substances. The samples of three colors (green from kale, white from Chinese cabbage, and red from tomato) were used to determine the effect on the assay at different concentrations of 25, 50, and 100 µg/kg compared with 10% methanol in PBS at a pH of 7.2. The results are shown in Table 1 and confirmed that color had an effect on the binding, but it did not make a significant difference. As color had an effect on the assay, pooled samples were prepared from those three kinds of vegetables with no chlorpyrifos residue after analyzing by GC-FPD. It is good to have used vegetables of different colors, but it does not imply the matrix effects are only due to color. The pooled samples were extracted and used as a matrix for a standard curve for calculating the concentration of chlorpyrifos in the samples. Due to the effect of the organic solvent and the matrix effect, the pooled samples were extracted by using the methods as previously reported. Spiked pooled control samples were extracted between every 20 unknown samples.

### 3.2. The Optimization of ic-ELISA for Detecting Chlorpyrifos

The optimization parameters are shown in Table 2. The recoveries of chlorpyrifos from the spiked pooled vegetable samples ranged from 95.3% to 117.8% (mean = 102.9%). The precision was reported by the coefficient of variation (CV), and the CV ranged from 4.6% to 6.7%, indicating that the developed ic-ELISA can be used for the detection of chlorpyrifos with good reproducibility. The sensitivity was evaluated by determining the limit of detection (LOD) and limit of quantification (LOQ) using the IC_20_ to IC_80_ (0.40 to 41.8 µg/kg). The LOD for the chlorpyrifos at the IC_15_ was 0.26 µg/kg. The LOD and LOQ values were lower than the MRLs established by Codex (Codex Alimentarius Commission, 2019) for vegetables samples. The ability of the developed ic-ELISA was determined: the positive sample was the detected sample by GC-FPD; the negative sample was not detected by GC-FPD; and the positive samples by this method and GC-FPD were 46 and 33 samples, respectively. The results show good correlation with the results from GC-FPD (R^2^ linear = 0.910), where a value of 0.954 was obtained by the Pearson correlation test. The results from ic-ELISA provided overestimated qualitative results, these maybe from the matrix effect and very high sensitivity of the ic-ELISA. The pretreatment of samples could improve the sensitivity because of less dilution [47]. The developed ic-ELISA shows good specificity to chlorpyrifos and chlorpyrifos-methyl, while the previous report showed broad specificity to a group of pesticides, making it unable to quantify a single pesticide within the same sample. This low cross-reactivity to other compounds in the group of OPs led to good specificity to chlorpyrifos and good accuracy to identify the residue in real samples. To validate the result from ic-ELISA by GC-FPD, there was thirteen false positive samples with no false negatives, so that could be applied as both a screening test and quantitative test with a calibration curve. The present study applied the developed ic-ELISA to quantify the single OP, i.e., chlorpyrifos, and the positive sample was confirmed by GC-FPD. The high concentrations above the limit of quantification were diluted and repeated. The concentrations of chlorpyrifos residues in vegetables were calculated against the standard curve (Figure 2) and reported in µg/kg of vegetable sample. The results of the immunoassay were usually validated with the gold standard technique, i.e., a chromatographic technique [34,40,48].

### 3.3. Monitoring of Chlorpyrifos Residue in Vegetable Samples from Northern Thailand

Currently, there are already several studies that indicate that ELISA can be used to analyze agricultural products and food samples after solvent extraction [40,49,50,51]. The developed ic-ELISA is very useful for analyzing samples in a high number: it is a simple process, using inexpensive equipment, and done in a rapid manner. The conventional methods, such as GC-FPD, used 50 min per sample (one run), while ic-ELISA can analyze 50 samples in one day in approximately 4 h. The developed ic-ELISA exhibited good accuracy and reproducibility, and it is ideally suited as a fast, high-throughput, and low-cost screening test for organophosphate (OP) residues prior to chromatographic analysis. The quality control of ic-ELISA was assessed by analyzing chlorpyrifos intra-batch (analyzing within group of samples, *n* = 5) and inter-batch (analyzing between groups of samples, *n* = 10). The results showed good intra-batch (%CV = 11.8) and inter-batch (%CV = 16.3), respectively. Table 3 reports the concentration of chlorpyrifos in vegetables surveyed from local markets in three provinces of northern Thailand in 2016, as well as comparison with EU-MRL for assessing the situation of residue chlorpyrifos in vegetable samples from northern Thailand. The frequency that chlorpyrifos residues was found in the vegetable samples was 33.8%, and the highest levels were found in cucumbers, coriander, and morning glory, at 145, 38, and 33 µg/kg, respectively. Among the compounds detected, chlorpyrifos was detected in all kinds of vegetable samples. The highest chlorpyrifos level was found in cucumber (275 µg/kg), followed by coriander (145 µg/kg) and Pakchoi (60.6 µg/kg). Previous studies reported chlorpyrifos residues in coriander, Chinese cabbage, broccoli, and Chinese kale [4,52], as well as in mock pakchoi [53], with equal concentrations and detection of chlorpyrifos in the same types of vegetables.

The findings showed that the median concentration of the detected chlorpyrifos for all the vegetables was above the maximum residue limits established by the European Union, except for Thai eggplant and broccoli. No Thai-MRL is yet established for chlorpyrifos residues in vegetables, i.e., water convolvulus, mock pakchoi, and yard long bean. It is concluded that awareness, safety education, and strict regulation of pesticide use is still necessary.

## 4. Conclusions

The present study shows the ic-ELISA developed from pAb exhibited reliable and high sensitivity for chlorpyrifos detection. In comparison with other ELISA approaches, this developed ELISA exhibited good resilience against organic solvents (10% methanol) and a large linear range that spans two orders of magnitude. The developed ELISA was suitable for different levels of chlorpyrifos in samples. The developed inhouse immunoassay method can be used for analyzing chlorpyrifos residue in a high number of vegetable samples, and it is rapid, inexpensive per sample, and can be used for a relevant diversity of samples. The data of chlorpyrifos residues in vegetable samples from the northern part of Thailand detected in many kinds of samples contained chlorpyrifos. Chlorpyrifos residues were the most frequently detected in coriander, with 10 samples (35.71%) exceeding the EU-MRL, while spring onion had the highest percent of detection (75%) exceeding the EU-MRL. A positive sample should be confirmed by standard analytical technique, such as GC-FPD. The contamination levels of chlorpyrifos residues could be considered a possible public health problem because it is the most popular insecticide used among farmers and one of the highest volumes of organophosphate insecticides imported into Thailand.

## Figures and Tables

**Figure 1 ijerph-17-04723-f001:**
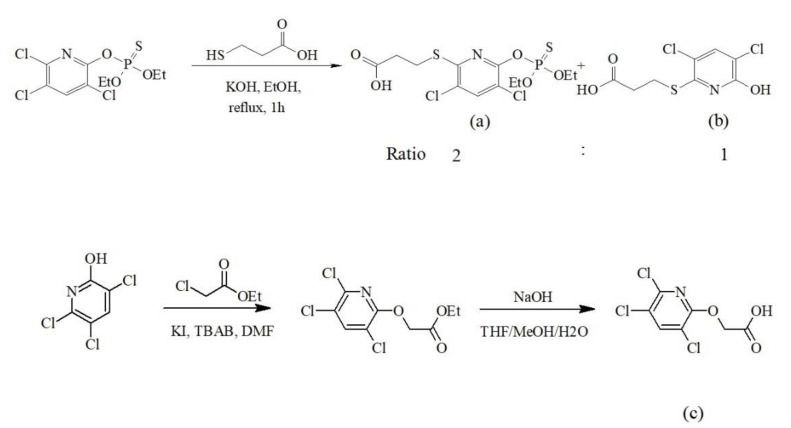
The synthesis of haptens was performed as previously described [32]. Compounds (**a**), (**b**), and (**c**) were separated by column chromatography.

**Figure 2 ijerph-17-04723-f002:**
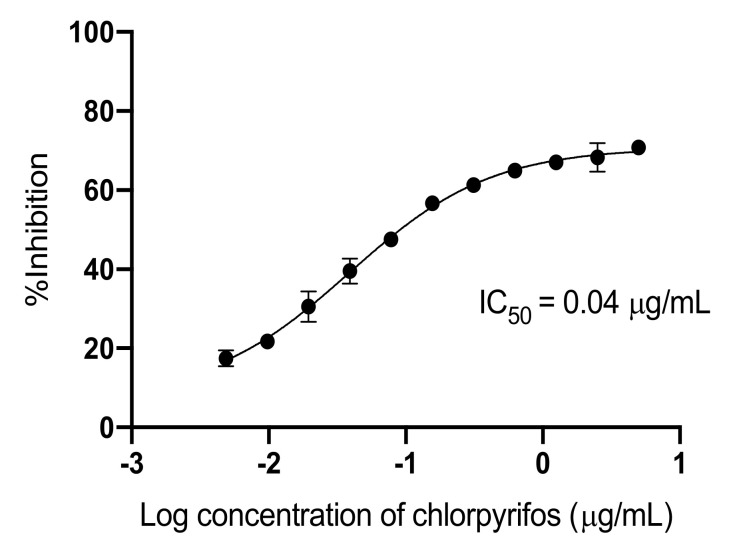
Standard curve of organophosphate pesticide chlorpyrifos by optimized ic-ELISA. The coating antigen was (c)-ova 1µg/ml, and the serum dilution was 1:1000.

**Table 1 ijerph-17-04723-t001:** Effects of the organic solvent, PBS, and matrix effect from the color of vegetables on ic-ELISA.

Methanol	IC_50_ (µg/kg)	PBS, pH = 7.2	IC_50_ (µg/kg)
50%	1.60 ± 0.60	1×	1.40 ± 0.68
25%	0.80 ± 0.12	2×	1.20 ± 0.56
10%	0.40 ± 0.24	3×	1.80 ± 0.62
5%	0.52 ± 0.33	4×	2.20 ± 1.02
		5×	4.00 ± 2.24
		DI water	10.0 ± 6.78
Matrix effect from the colors of vegetable samples		IC_50_ (µg/kg)	
Tomato		0.90 ± 0.42	
Kale		0.90 ± 0.38	
Chinese cabbage		1.40 ±0.66	

**Table 2 ijerph-17-04723-t002:** Accuracy, precision, limit of detection (LOD), and limit of quantification (LOQ) of the developed ic-ELISA.

Parameters	ic-ELISA	GC-FPD
LOD, µg/kg	0.26 (IC_15_)	1.00
LOQ, µg/kg	0.40–41.8 (IC_20_–IC_80_)	2.00
Precision (%CV, *n =* 4)		
2.00 µg/kg	4.60	2.00
1.00 µg/kg	6.70	2.58
0.50 µg/kg	5.50	3.40
Accuracy (%Recovery: *n =* 5)		
2.00 µg/kg	95.3	114.32
1.00 µg/kg	117.8	99.42
0.50 µg/kg	95.72	97.60

**Table 3 ijerph-17-04723-t003:** The chlorpyrifos residue in vegetable samples from northern Thailand.

Vegetables Samples	Number of Samples	Chlorpyrifos Residue (µg/kg)		Median	Min–Max	Maximum Residue Limit (µg /kg)		
		Positive Number	Mean ± SD			Thailand ^(1)^	Codex ^(2)^	EU ^(3)^
Coriander	28	14	38.3 ± 38.4	24.05	3.60–145	-	-	10
Yard Long Bean	23	10	16.6 ± 15.5	12.35	3.20–56.0	-	-	10
Pakchoi	17	3	30.8 ± 27.8	26.30	5.60–60.6	-	-	10
Cabbage	17	2	13.4 ± 0.01	13.45	9.00–17.9	-	100	10
Chinese Cabbage	13	1	332 ± 0.00	332.3	332	-	100	10
Morning glory	13	3	32.8 ± 2.14	31.7	31.5–35.3	-	-	10
Cauliflower	9	3	16.4 ± 9.05	13.00	9.60–26.7	-	50.0	50
Spring onion	8	6	18.6 ± 8.97	16.45	6.10–29.6	-	200	10
Broccoli	7	2	9.00 ± 0.42	9.00	8.70–9.30	-	2,000	10
Chili	7	3	24.5 ± 8.25	23.0	17.10–33.4	3000	-	10
Thai Eggplant	6	1	6.20 ± 0.00	6.20	6.20	200	100	10
Chinese Kale	5	2	30.0 ± 23.8	26.95	10.1–43.8	-	-	10
Cucumber	4	2	146 ± 183	146.3	16.9–275	-	-	10
Tomato	3	2	23.5 ± 7.50	23.50	18.2–28.8	-	2000	10
Total	160	54	35.3 ± 58.4	18.40	3.2–332	-	-	-

Note: ^(1)^ Thai Food and Drug Administration (2017), ^(2)^ Codex Alimentarius Commission (2019), ^(3)^ European Commission (2020).

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
