# Peer review of "Monitoring of the Organophosphate Pesticide Chlorpyrifos in Vegetable Samples from Local Markets in Northern Thailand by Developed Immunoassay"

_ijerph, 2020, doi:10.3390/ijerph17134723_

Round 1
Reviewer 1 Report
The article entitled “Monitoring of the Organophosphate Pesticide Chlorpyrifos in Vegetable Samples from Local Markets in Northern Thailand by Developed Immunoassay” is suitable for publication in this magazine. The quality of the manuscript is adequate. Next, the strengths of this article are detailed, for which I consider that it should be published without revisions.
1) The article deals with the evaluation of the content of chlorpyrifos, an organophosphate pesticide that is widely used for crops. The importance of this article is that this substance is toxic and chronic exposure to it has been linked to neurological effects, developmental disorders and autoimmune disorders. In addition, it should be noted that, due to its high persistence, this substance has been banned in countries such as Germany, Denmark or Sweden. Currently, the European Union has banned its use in January this year. However, countries like Thailand continue to use it, and it is necessary to monitor its levels.
2) This study has carried out a monitoring of the levels of chlorpyrifos in plant samples from local markets in areas of Thailand, with the aim of studying the concentrations that are capable of containing vegetables intended for human consumption. It is, therefore, a job of great interest within the area of ​​Food Security and Risk Assessment.
3) The method used for the determination of chlorpyrifos in vegetables has been the ic-ELISA with GC-FPD. The authors have demonstrated the efficacy of the method as well as the accuracy.
4) This work has determined that cucumbers have the highest levels of chlorpyrifos. This study is of great interest for risk assessment as it has presented new data from agricultural areas in Thailand.
5) Likewise, the authors have used a method that has proven to be adequate for this determination, and this method may be implemented by other authors who access this study.
Author Response
Thank you so much for the valuable comments from the reviewers. The comments are helpful to improve our manuscript. The point-by-point replies are attached.

Reviewer 2 Report
This manuscript reports the development of an enzyme-linked immunosorbent assay (ELISA) for chlorpyrifos using a polyclonal antibody raised to hapten conjugates previously reported in the literature. Whilst the development of new convenient approaches to the sensitive quantification of chlorpyrifos are of clear interest to analytical chemists in the environmental and public health areas, I feel that the authors have not adequately explained the advantages of their method that differentiate it from the existing set of reported enzyme immunoassays for chlorpyrifos. Furthermore, there are a number of deficiencies in the description and analysis of the assay results and these should also be addressed. More specific details are provided below.
- The authors should clarify the novelty of their work and differentiate it from the existing literature. What particular benefits does your analytical approach have over other options?
- The manuscript has numerous spelling and typographical errors and should be carefully edited.
- Page 1 lines 27-28 “Chlorpyrifos was detected . . . at 33.8% . . .” What do you actually mean here? Are you saying that 33.8% of samples had detectable chlorpyrifos? Clarify this in the manuscript.
- Page 1 lines 28-31 “. . . with the highest . . . 60.6 µg/kg).” This is contradictory. You have quoted two different figures for highest concentration for cucumber and coriander. You need to clarify what the highest concentrations detected were for each sample and resolve this contradictory description.
- Why has no 13C NMR data been provided for hapten c?
- Page 4 lines 130-131 “. . . based on mass spectra of ultraviolet spectra . . .” This statement does not make sense. In the manuscript, clarify the roles played by the UV-visible spectra and the mass spectra.
- Page 4 lines 146-147 “. . . incomplete Freund’s complete adjuvant . . .” You need to clarify which adjuvant you are using, complete or incomplete.
- Page 4 lines 153-154 “Pooled serum from . . . for further experiments.” Here you suggest that heart blood was taken after the third immunisation but earlier you state it was taken after the sixth immunisation. Clarify the timing of the heart blood sampling.
- Page 4 lines 163-164 “. . . concentrations of PBS . . . DI water.” You should clarify in the manuscript the molarity of the PBS used at 1 x concentration. Also, why are you using PBS for this test but PBS-T for the serum dilution in the assay?
- Page 4 line 173 “. . . of serum diluted . . . 1:1000) . . .” Has this dilution factor been optimised? If so, then provide details in the manuscript.
- Page 5 lines 175-176 “. . . goat anti-mouse . . . in PBS . . .” If the secondary antibody has no enzyme attached then how are you getting a colour change in the ELISA? Modify the manuscript to explain this.
- Page 5 line 178. The pH of the citrate-acetate buffer is missing and needs to be added.
- Page 5 lines 197-198 “To evaluate the . . . pesticide chlorpyrifos . . .” This statement does not make sense. Surely you are trying to evaluate the performance of your assay not the performance of chlorpyrifos as a pesticide.
- Page 6 lines 246-247 “. . . titer of antibody . . . was equal.” So, are you saying that the serum antibody titre after the third immunisation was the same as the heart blood sample? Change the wording to provide clarification of this in the manuscript.
- Page 7 lines 279-280 “The pooled samples . . . in samples.” If you are using this as a matrix for the standard curve, then how do you know that this pooled sample does not have chlorpyrifos present? In other words, how do you know that this is a true blank?
- Page 7 lines 287-288 “The accuracy was . . . variation (CV) . . .” CV is not a measure of accuracy, it is a measure of precision.
- Page 7 lines 293-300 “The ability of . . . QuEChERS.” Based on the data reported here, the assay tends to suffer from significant rate of false positives. This is consistent with an overestimate of the chlorpyrifos concentrations not the underestimate you mention in the text. Also, as an explanation for this false positive rate, you suggest that it may be caused by the assay detecting chlorpyrifos-methyl, but you have stated that chlorpyrifos-methyl has no reported use in Thailand. It would therefore appear that the explanation for these data lies elsewhere.
- Page 7 lines 305-306 “To validate the . . . false negatives . . .” This statement directly contradicts the data given earlier in this section where the ELISA gave 46 positives and the GC gave 33 positives, i.e. 13 assay false positives. The authors need to provide consistent and clear data on this and with appropriate accompanying commentary.
- For all assessments of “positive” samples and “negative samples” the authors should clarify the concentration cut-off used. Is this the MRL?
- Page 7 line 308 “. . . qualitative test with a calibration curve . . .” If you have a calibration curve then you in fact have a quantitative assay, not just a qualitative test.
- The calibration curve for the assay needs to be given in the manuscript in one of the figures.
- Table 2. How was the % recovery computed? As these concentrations were obtained by spiking into vegetable extract, then how did you obtain a blank background matrix? These details need to be added to the manuscript.
- Page 8 line 324 “. . . intra-batch . . . inter-batch . . . reproducibility.” Define what constitutes a “batch” in this assessment of the reproducibility.
- Page 10 lines 345-346 “. . . in coriander . . . EU-MRL . . .” The data in Table 3 clearly show that the maximum concentration found was 145 µg/kg which is clearly much higher than the EU-MRL of 10 µg/kg, so how can you state in the conclusions that no coriander samples had concentrations exceeding the EU-MRL?
- Page 10 line 346 “. . . spring onion . . . EU-MRL.” Don’t you mean the highest % of samples above the EU-MRL, not the highest number of samples, as coriander and yard long bean gave higher positive numbers.
Author Response
Thank you so much for the valuable comments from the reviewers. The comments are helpful to improve our manuscript. The point-by-point replies as attached

Reviewer 3 Report
Lines 37 and 38 – The sentence “on controlling different kinds of pests, including termites, mosquitoes, and round worms, because chlorpyrifos is a broad-spectrum organophosphorus insecticide” should be changed as chlorpyrifos is mentioned as broad-spectrum organophosphorus insecticide and round worms are not insects;
Line 39 – The word” however” should be replaced by another such as additionally; chlorpyrifos use causes environmental problems as well as food security problems.
Line 41 – using the product in large volumes is not related to the diversity of samples/foods in which can be detected; please revise the sentence.
Line 43 – since the damage to the central nervous system is relevantly due to the inhibition of cholinesterase in the brain, the authors could mention it; it seems more important than the mention to plasma cholinesterase inhibition.
Line 227 – If there are more methods to quantify chlorpyrifos, the authors could explain the reason(s) for choosing Gas chromatography-flame photometric detection to compare the immunoassay results; basically explain briefly why is Gas chromatography-flame photometric detection a reference method or add a reference.
Line 316 – In the objectives the author should mention also the assessment of chlorpyrifos residues in food samples, even if only as a preliminary study, in a perspective of food security; in fact the values found are being compared with reference values.
Line 342 – It could also be mentioned that the method can be used in a relevant diversity of samples.
Author Response
Thank you so much for the valuable comments from the reviewers. The comments are helpful to improve our manuscript. The point-by-point replies as attached.

Reviewer 4 Report
The paper should be improved in particular in the result and in the conclusion because the results indicate that the conclusions are different from the text in the conclusion. The median values found are higher that the EU residue limit for most of the vegetables.
lines 82-88 2.2 Standard Organophosphate pesticides - the purity of the standard used should be added. How the authors prepare the standard solution? Where and how the standard are stored?
The author should explain better that the method can be applied for monitoring scope and that the positive sample should be confirmed by other analytical technique.
Author Response

(The authors gave the same response as above.)

Round 2
Reviewer 2 Report
The authors have made a number of revisions to the original manuscript that address most of the individual points raised in my earlier review. However, there are a significant number of those points that have not been adequately addressed, including in my opinion, the core issue of the authors not adequately defining the key advantages and different features of their assay in comparison to existing ELISA technology. More detailed comments on matters that the authors still should address are given below.
- The extra statements added by the authors concerning novelty of their assay are in fact simply generic descriptions of the features and benefits of ELISAs in general in comparison to other instrumental methods such as chromatography. What is needed is a clear description in the manuscript text of the points-of-difference between the current assay and other ELISA approaches, e.g. with regards to assay format, chemistry, types of samples analysed and their associated background matrices, sensitivity, precision etc.
- The manuscript still has a significant number of typographical and spelling errors and so needs more thorough editing.
- Page 1 lines 27-28 “Chlorpyrifos was detected . . . at 33.8% . . .” What do you actually mean here? Are you saying that 33.8% of samples had detectable chlorpyrifos? Clarify this in the manuscript. The authors have given an adequate clarification of this in the comments to the reviewer but have made no change in the text of the abstract. It is important that the abstract text is clear and unambiguous.
- Page 1 lines 29-31 “The highest median . . . 60.6 ug/kg).” These median figures do not match those given in Table 3, so what exactly do you mean by “highest median” in this context? Provide clarification in the manuscript text.
- Page 4 lines 136-137 “. . . based on mass spectra of ultraviolet spectra . . .” This statement does not make sense. In the manuscript, clarify the roles played by the UV-visible spectra and the mass spectra. The authors have given an adequate clarification of this in the comments to the reviewer but have made no change in the manuscript text.
- Page 4 lines 158-160 “Pooled serum from . . . for further experiments.” Here you suggest that heart blood was taken after the third immunisation but earlier you state it was taken after the sixth immunisation. Clarify the timing of the heart blood sampling. Because you are still reporting two different timings for taking heart blood, there is still confusion over which sampling is used for which purpose. This needs to be carefully clarified in the manuscript text in section 2.3.3.
- Page 6 lines 253-256 “The polyclonal antibody. . . vegetable samples.” It is still unclear what the titre in serum samples was equal to. This should be clarified in the manuscript text. Also, if the polyclonal antibody is taken from pooled serum just after the third immunization, then why was blood collected after the sixth immunisation for three days and what was it used for (see comment 6 above)?
- Table 2. How was the % recovery computed? As these concentrations were obtained by spiking into vegetable extract, then how did you obtain a blank background matrix? These details need to be added to the manuscript. These details were provided in the response to the reviewer but not in the manuscript text.
Author Response
Thank you so much for the valuable comments from you. The comments are helpful to improve our manuscript. The point-by-point replies as attached

Reviewer 4 Report
Thank you for the answer. The paper is acceptable.
Author Response
Dear reviewer,
Thank you very much for your review and accepted our manuscript.
Sincerely yours,
Surat Hongsibsong
Round 3
Reviewer 2 Report
The authors have made minimally adequate changes to the manuscript in response to my earlier review comments.